# A New Perspective on Doubly Special Relativity

**J. M. Carmona** [1,2] , **J. L. Cortés** [1,2] , **J. J. Relancio** [2,3,*] and **M. A. Reyes** [1,2]

1   Departamento de Física Teórica, Universidad de Zaragoza, 50009 Zaragoza, Spain
2   Centro de Astropartículas y Física de Altas Energías (CAPA), Universidad de Zaragoza, 50009 Zaragoza, Spain
3   Departamento de Matemáticas y Computación, Universidad de Burgos, 09001 Burgos, Spain
*   Correspondence: jjrelancio@ubu.es

**Abstract:** Doubly special relativity considers a deformation of the special relativistic kinematics parametrized by a high-energy scale, in such a way that it preserves a relativity principle. When this deformation is assumed to be applied to any interaction between particles, one faces some inconsistencies. In order to avoid them, we propose a new perspective where the deformation affects only the interactions between elementary particles. A consequence of this proposal is that the deformation cannot modify the special relativistic energy–momentum relation of a particle.

**Keywords:** quantum gravity; doubly special relativity; relative locality; relativistic deformed kinematics

## 1. Introduction

The aim of this work is to present a new proposal to introduce a departure from the notions of spacetime and locality in special relativity (SR). The first reference to the possible limitations of these notions goes back to Einstein himself (1905) [1]: *"We shall not here discuss the inexactitude which lurks in the concept of simultaneity of two events at approximately the same place, which can only be removed by an abstraction".*

This departure might be very relevant in the development of theories going beyond SR trying to put in the same scheme general relativity (GR) and quantum field theory (QFT). Indeed, a possible source of inconsistencies that impedes the unification of these two theories is the role that spacetime plays in them. While in QFT spacetime is given as a rigid static framework in which propagation and interactions can be described, in GR spacetime is understood as a dynamic deformation of the Minkowski spacetime modeled by energy–momentum sources. Both theories are expected to be limiting cases of a quantum gravity theory (QGT), where spacetime would reveal its quantum nature, leading to a completely new structure. This is the case of some proposals for a QGT, such as string theory [2–4], loop quantum gravity [5,6], causal dynamical triangulations [7], or causal set theory [8–10]. However, the correct fundamental spacetime structure is still unknown to date.

An alternative, bottom-up, approach to QGT may come from trying to incorporate some generic expected modifications of the classical spacetime to our present, low-energy theories, as a way to try to capture residual quantum gravity effects. Such modifications include, most notably, departures from the symmetries and from the standard notion of locality of SR and can indeed have a well-defined and testable phenomenology [11]. Two main possibilities have emerged in this quantum spacetime phenomenology [12], affecting the standard kinematics of SR in a different way: either Lorentz invariance is broken, which implies the existence of a privileged system of reference [13,14], or the symmetries of SR are deformed, as in the so-called "Doubly Special Relativity" (DSR) scenario, so that a relativity principle is still present [15,16].

DSR is an attractive theoretical possibility, since it represents a step from SR which is analogous to the one that SR made from Galilean relativity, and it dismisses the need

of a privileged system of reference. The kinematics of DSR models is characterized by two different ingredients that may appear in a generic deformation: a modification of the dispersion relation of SR (MDR) and a modification in the composition law of momenta (MCL), which is no longer additive, therefore affecting the conservation of energy and momentum in processes. The MDR adds new terms to the quadratic relation between the energy and momentum of a particle in SR, which are proportional to powers of the energy of the particle divided by a high-energy scale $\Lambda$, usually considered as the Planck energy. The MCL for two particles differs from the simple addition in terms involving products of momenta of both particles and again, the inverse of the high-energy scale $\Lambda$, since we want to recover SR when this scale goes to infinity. While an MDR may be absent in specific DSR models (so that the dispersion relation is that of SR), an MCL is always present, as well as deformed Lorentz transformations, in order to keep the relativity principle [17]. The MCL can be related [18] to the coproduct operation in the formalism of Hopf algebras [19], being the $\kappa$-Poincaré example [20,21] the most studied model. In the particular case of $\kappa$-Poincaré, the MCL is noncommutative but associative, so that the composition law of a multiparticle system is completely defined by gathering momenta in pairs.

However, the implementation of the deformed kinematics of DSR leads to several problems of consistency of the theory. In particular, a deformation of the usual relativistic expression of the dispersion relation leads to the so-called "soccer-ball" problem [22,23]: the terms proportional to the inverse of the high-energy scale (even when considering a Planckian energy) produce a huge contribution when considering a macroscopic object. This problem is not exclusive to the MDR; it also affects the MCL, whose nonlinear terms would give enormous contributions to the total energy of a macroscopic object, when expressed in terms of the energy of its constituents. Although some proposals have been pointed out to solve this problem [23–25], they are not free from issues, as we discuss in Section 2.

Another consistency problem of DSR has to do with the apparent nonlocal influences that a nonadditive composition law seems to suggest [26]. This "spectator problem" is more evident when one tries to analyze the translational symmetry of a process that includes several interactions [27–29]. The generator of translations, the total momentum defined with the MCL, contains momenta of particles that do not directly participate in one of the interactions, but that appear in other interactions that form part of the process. The translational invariance can then only be consistently implemented if one knows the whole sequence of causally connected vertices for the particles involved in every interaction. This means that a complete knowledge of the content of the Universe should be given for the description of every process, which impedes us having a physical description of reality.

In this work, we present a solution to these problems of consistency by introducing a new perspective on how the deformed kinematics of DSR plays a role in processes of particles. Essentially, we propose that DSR should be seen as a way to go beyond the standard locality of interactions present in relativistic quantum field theories that is still compatible with relativistic invariance. Our proposal is described in Section 2 and, in particular, it leads us to conclude that modified composition laws should appear only whenever elementary particles are involved in an interaction, while, at the same time, their dispersion relation should be the same as in special relativity. As we see in Section 3, this new perspective on DSR has specific phenomenological implications that can be distinguished from those considered in the standard interpretation of DSR. We end with a brief summary in Section 4 and give some technical details on a modified composition law compatible with the present proposal in Appendix A.

## 2. DSR as a Way to Go Beyond Local Interactions

We present in this work a new interpretation of DSR as a way to go beyond local interactions in a quantum relativistic theory. In relativistic quantum field theory (RQFT), interactions are local in the sense that they are defined by products of more than two fields in a spacetime point, which is the way one introduces interactions in a Lagrangian density

of the theory of quantum fields. When one uses the plane-wave expansion for the fields, the locality of the interactions automatically leads to linear relations between the momenta defining the plane-wave expansion when one considers the integral over spacetime of the Lagrangian density (action).

One can identify DSR with a nonlinear modification of the sum of momenta (composition of momenta) depending on an energy scale $\Lambda$. This can be implemented in the field theory framework as a deformation of the product of fields (see [30] and references therein). However, this mathematical formalism has not yet provided a systematic treatment of the effects of the deformation in the calculation of observables in particle processes. In any case, what is clear is that the nonlinear relations between momenta that are obtained through that procedure imply a loss of the locality of interactions. Interactions in DSR do not define a point in spacetime but a region of size $\ell \sim (1/\Lambda)$. This is the ingredient that we take as the key point for the proposed new interpretation of DSR.

We have explored nature at the microscopic level up to scales of the order of $(\text{TeV})^{-1}$ with collisions of particles in the highest energy accelerators. We have not seen any sign of deviations from the locality of interactions; then, we conclude that if DSR is realized in nature, the energy scale $\Lambda$ of DSR should be larger than 1 TeV.

The interactions involving a system whose size is much larger than the scale $\ell$ are not be affected by the transition from SR to DSR. This argument reduces the search for observable effects of DSR to the interactions of what we identify at present as elementary particles (leptons, quarks and mediators of the interactions in the standard model of particle physics).

The above perspective of DSR can be contrasted with the standard interpretation in which DSR is identified as the necessary modification in SR to make relativistic invariance compatible with a (minimum) length scale. One finds that the linear Lorentz transformations between different frames in SR have to be replaced by nonlinear transformations. This leads to a deformation of the energy–momentum relation of a particle in SR and also to a deformation of the relations imposed by the energy–momentum conservation associated with the translational invariance. When one considers a classical model for the interaction of particles based on the deformed energy–momentum conservation, one finds that the notion of locality of SR (interactions associated to the intersection of the worldlines of the particles) becomes an observer-dependent property; absolute locality is replaced by relative locality [31,32]. The deviations from locality depend on the observer when one considers a family of observers related by translations.

The search for observable effects in the standard interpretation of DSR follows different rules from those of the proposal in this work. One usually interprets the deformation of SR as a signal of a theory of quantum gravity, identifying the energy scale of the deformation $\Lambda$ with the Planck scale. The limitation on the energies one can reach in accelerators, or even in high-energy astrophysics, makes any effect of DSR in the interaction of particles unobservable. One has to look for the amplification of the effects of DSR in the propagation of particles over astrophysical distances due to the modification of the energy–momentum relation and, correspondingly, of the velocity of propagation. If one changes the choice of energy–momentum variables, one will have a different energy dependence of the velocity of propagation; in fact, one can always choose energy–momentum variables to make the velocity of propagation energy-independent. In this case, however, translations generated by these variables act nontrivially in spacetime and, when one compares two observers separated by astrophysical distances, one finds once more the amplification of the effects of DSR [33].

In the new perspective of DSR as a small departure from the locality of interactions, there are no effects of DSR in the propagation of a free particle, and then no amplification when one considers astrophysical distances. The only way to find observable effects is to assume that one can have interactions of particles with energies approaching the scale $\Lambda$ of the deformation.

In order to illustrate the differences between the standard interpretation of DSR and the new perspective proposed in this work, we consider a process with two interactions in regions of spacetime separated by a large distance.

### 2.1. Production, Propagation and Detection of a Very High Energy Particle

We consider a process with an interaction producing a particle which propagates and is detected by a second interaction, as shown in Figure 1.

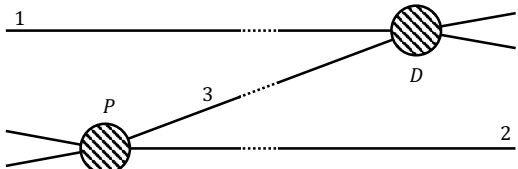

**Figure 1.** A particle, labelled '3', is produced in an interaction ($P$), propagates over a large distance, and is detected in a second interaction ($D$). Particles labelled by '1' and '2' only participate in the interactions $D$ and $P$, respectively.

We assume that each interaction is an elastic scattering of two particles. From a kinematic point of view, the interaction produces a change of the momenta of the particles. This change of momenta is determined in SR by a conservation law. The effect of DSR is to modify this conservation law through a deformed composition of momenta and then, to change the possible momenta of the particles in the final state of the process with respect to their values in SR. The relativistic invariance of the deformed kinematics generally requires a modification of the energy–momentum relation of the particles compatible with the modification of the composition of momenta, since both modifications are related by the so-called "golden rules" [28,34]. However, there are examples of modifications of the composition of momenta compatible with the energy–momentum relation of SR, such as the classical basis [35] of $\kappa$-Poincaré and the well-known Snyder kinematics [36].

In previous works [27–29] on DSR, the kinematics of the process in Figure 1 was studied based on a modified implementation of the translational symmetry. One assumed the conservation of the total momentum of the three-particle system defined in terms of the modified composition of their momenta, which could be taken before the first interaction, between the two interactions, or after the second interaction. One could consider that the exchange of momenta in each interaction involves the momenta of three particles. In fact, in the algebraic interpretation of DSR, based on a $\kappa$-deformed Poincaré Hopf algebra, the composition of momenta is determined by the coproduct of momentum operators, which makes no reference to spacetime and then, the momentum of particle 1 could be modified by the interaction $P$ even if this particle is far away from the spacetime region where the interaction takes place [26]. Even if one assumes that this is not the case, nothing tells us that the conservation of the total momentum of the three-particle system before and after the interaction $P$ will lead to a relation between momenta independent of the momentum of particle 1. The total momentum of the three-particle system before the first interaction will be the result of the composition of the momentum of particle 1 and the momentum variables of the other two particles participating in the interaction $P$ whose momenta change due to the interaction. However, nothing fixes the ordering of the momentum variables in the expression of the total momentum before and after the interaction. In particular, one could have a case where the conservation of the total momentum can be expressed as

$$p^{(1)} \oplus p^{(2)} \oplus p^{(3)} \;=\; p^{(2)'} \oplus p^{(1)} \oplus p^{(3)'}, \tag{1}$$

where the symbol $\oplus$ denotes the modified composition between momenta. An example of such MCL is given in Appendix A. The relation between the variables $(p^{(2)}, p^{(3)})$ and $(p^{(2)'}, p^{(3)'})$ depends on $p^{(1)}$ in this case.

The same applies to the transition due to the detection of the particle, which can depend on the momentum of particle 2. In this case, one cannot treat both interactions independently, even if the produced and detected particle is propagating over a very large distance. Therefore, DSR leads to a violation of the cluster property of interactions, which is at the basis of RQFT [37].

The previous argument also raises doubts on the consistency of considering the process in Figure 1 as an isolated system. The interactions producing particle 1 and the two particles in the initial state of the interaction $P$ could affect the kinematics of the process and should be included. One could even go further and consider a fourth particle which does not participate in any of the two interactions. The conservation of the total momentum of the four-particle system, including this new particle, would lead to an exchange of momenta between the particles depending on the momentum of the fourth particle. This is what is known as the spectator problem [27–29].

The main idea in the new perspective of DSR is to associate the deformation to a deviation from the locality of the interactions characterized by a length scale $\ell \sim (1/\Lambda)$, where $\Lambda$ is the energy scale of DSR. When one considers an interaction of particles which are seen as elementary when explored at distances larger than the scale $\ell$, one sees an effect of the deviation from the locality of interactions; the change of momenta due to the interaction is determined by the kinematics of DSR. On the other hand, when one considers an interaction with composite particles whose size is much larger than the scale $\ell$, one can neglect the deviations from locality, and the change of momenta produced by the interaction is determined by the kinematics of SR.

Let us assume that in Figure 1, particles 2 and 3 are elementary but particle 1 is a composite particle, so that the interaction producing particle 3 is between elementary particles, and the interaction in its detection involves a composite particle. A possible relation for the change of momenta in the interaction $D$ is

$$p^{(1)} - p^{(1)'} \;=\; p^{(3)'} \oplus \hat{p}^{(3)}\,, \qquad (2)$$

where $\hat{p}^{(3)}$ is the antipode (see Appendix A) of the momentum of particle 3 before its detection. We selected in Equation (2) one of the two possible orderings of the composition of momentum variables on the right-hand side. The left-hand side is a difference of momentum variables of the composite particle 1, whose kinematics is not affected by the deformation of SR. Then, they transform linearly under Lorentz transformations, and this implies that the composition of momentum variables on the right-hand side also has to transform linearly.

The assumption that one can have interactions with different kinematics eliminates the arbitrariness in DSR associated with the different choices of energy–momentum variables. The composition of two momentum variables as well as a single momentum variable have to transform linearly under Lorentz transformations. Therefore, there is no modification of the dispersion relation with respect to SR and there is no signal of the deformation in the propagation of a particle. This agrees with the interpretation of DSR as a modification of the interaction between elementary particles instead of an effect in the propagation of particles in a quantum spacetime.

### 2.2. Deviation from Locality in DSR

Another consequence of the standard interpretation of DSR as a modified implementation of the translational symmetry generated by the total momentum of the three-particle system is that there is no observer that sees the two interactions in Figure 1 as local. Different observers related by a translation have a different origin of spacetime. An observer whose origin is close to the region where the interaction $P$ is produced sees this interaction as approximately local but sees deviations from locality in the interaction $D$ proportional to the distance between the regions of the two interactions. The same happens for an observer whose origin is close to the interaction $D$, who sees a deviation from locality in the interaction $P$ proportional to the distance from it. One refers to this property as relative

locality [31,32], which defines DSR in the sense that it reflects the effect in spacetime of the modification of the composition of momenta. In the standard interpretation of DSR, where one cannot treat the two interactions independently, the idea of relative locality plays an important role to understand that there is no conflict between the deviations from locality in DSR and the very precise experimental tests of locality [38].

In the new perspective of DSR proposed in this work, where one can consider the two interactions independently, one can directly use in all experimental tests of locality an observer whose origin is within the region where the interaction that detects the particle takes place. One can ignore the interaction $P$ producing the particle, and one does not need to refer to the idea of relative locality in order to see that there is no conflict between DSR and the very precise tests of locality.

The standard interpretation of DSR is formulated in the framework of a classical model of interaction of particles based on worldlines, while in the new perspective proposed in this work, the deformation is introduced in the relation between the momenta associated with a product of quantum fields. Due to this difference of frameworks used in the formulation of the transition from SR to DSR, one has a different perspective on the loss of the notion of absolute locality present in SR.

### 2.3. Solution of the Spectator and Soccer-Ball Problems in the New Perspective of DSR

If one associates the effects of DSR to a deviation from the locality of the interactions between elementary particles, it is natural to determine the kinematics of each interaction by the momenta of the particles participating in it. This means that the change of momenta due to the interaction $P$ is independent of the momentum of particle 1, which does not participate in the interaction. The same argument can be applied to the change of momenta due to the interaction $D$, leading us to conclude that it is independent of the momentum of particle 2. Then, both interactions can be treated independently, according to the cluster property, and each of them can be treated as taking place in an isolated system. As a consequence, the new perspective of DSR solves the spectator problem and treats the process as three independent steps: an interaction $P$ producing particle 3, the propagation of the free particle 3 and an interaction $D$ where particle 3 is detected, just as in the case of SR.

Another potential inconsistency of the original proposal of DSR comes when one considers the kinematics of macroscopic systems. For a microscopic system in DSR, one can understand the small deviations from SR as a consequence of the small ratio of the energies of the particles and the energy scale which parametrizes DSR. Then, however, if the modified composition of momenta is the same for any particle, including a macroscopic system, one would have very large corrections to the kinematics of SR, in obvious conflict with our observations at the macroscopic level. The solution that has been proposed for this paradox (soccer-ball problem [22,23,25]) is that the scale of energy which parametrizes the modification of the kinematics of a macroscopic system is much larger than the energy scale of DSR at the microscopic level. More specifically, one argues that the scale of deformation is proportional to the number of constituents of the macroscopic object by considering an approximation where those constituents all move with the same velocity [23]. Identifying those constituents with the atoms of a macroscopic object, one finds an effect on the kinematics of the macroscopic system of the same order as the correction to the kinematics of the atoms. The smallness of the energy of each atom compared with the energy scale of DSR solves the soccer-ball problem.

The previous argument can be criticized from different perspectives. Why should we identify the atoms as the constituents of the macroscopic system instead of the elementary particles (electrons, quarks and gluons)? Were we to identify these elementary particles as "constituents", how many of them would there be in, for example, a proton? How good is the approximation to consider all the constituents as a rigid system?

In the new perspective of DSR proposed in this work, a macroscopic system is a composite system with a size much larger than the scale $\ell \sim (1/\Lambda)$ of the deviation

from the locality of the interactions of elementary particles. Then, the interactions of macroscopic systems are not affected by DSR and, according to the relativity principle, the energy–momentum relation of SR applies to a macroscopic system. The soccer-ball problem is solved as a direct consequence of the assumption that DSR only applies to the interactions of elementary particles.

We note that there is some similarity between this solution to the soccer-ball problem and the one offered in ref. [24], where this "problem" is seen as a "case of mistaken identity", distinguishing between the composition law, which is relevant in the modification of interactions in a noncommutative spacetime (then modifying the standard notion of locality) and the (additive) total momentum of a multiparticle system. However, ref. [24] does not discuss how one should consider the interaction between macroscopic particles; in the present proposal, we deduce that they should be described by a standard composition law between momenta.

### 3. Observable Effects of DSR

In contrast to the case of the Lorentz invariance violation (LIV), where one can have observable effects of the deviation from SR in the kinematics of processes at energies much smaller than the energy scale of the LIV [11,13,14], the relativistic invariance of the deformation of the kinematics of SR leads to a suppression of any effect in the kinematics of a process by powers of the ratio of the energy of the particles and the energy scale of DSR. If this scale is of the order of the Planck scale, as usually assumed when one considers the quantum structure of spacetime as the origin of the deformation of the kinematics, the possibilities to observe the effects of DSR are reduced to the problematic access to the details of the initial state of the Universe or the final stages of the evaporation of a black hole.

A completely different phenomenology of DSR is based on time delays of massless particles. In this case, even if the modification in the velocity of propagation of a particle with respect to SR is proportional to the ratio of the energy of the particle and the energy scale of DSR, there is an amplification effect, proportional to the long distance that astroparticles travel from the source to our detectors on Earth, which could be measured by current experiments. Therefore, a lot of papers were devoted to this possible effect in the context of DSR [33,39–45].

In the perspective of DSR discussed in this work, since the dispersion relation is the one of SR, time delays are absent [33,42]. This means that the strong constraints on the high-energy scale based on the possibility of time delays [46–52] do not apply in our scheme.

This opens up an attractive alternative from a phenomenological perspective, by keeping an open mind about the value of the energy scale of DSR. From this point of view, one can see which bounds are on such scale from the lack of signals of a departure from the prediction of SR in the kinematics of processes of particles, and which places are best to look for a first signal of DSR. The best candidates are the interactions involved in very high energy astroparticle physics [53,54]. Since the effects of the deformation are restricted to the interactions of elementary particles in the new perspective of DSR proposed in this work, one is led to identify the observations of the elementary cosmic messengers (gamma rays and neutrinos) at the highest energies as the best window to look for observable effects of DSR. A difficulty one faces in this program are the uncertainties in the predictions of SR due to the limited knowledge of the astrophysical processes in which those messengers are involved, which have to be disentangled from the possible effects of DSR on the interactions of those messengers affecting their observation at the highest energies.

Another alternative is to look for the interactions of particles at the highest energies in laboratory experiments, where one is free from the astrophysics uncertainties, but the energies one can reach are lower, and then one has to identify a smaller effect. This leads to concentrate on the most precise observations (Z-line shape is a good example) or on the experiments involving the highest possible energies (Large Hadron Collider at CERN and a future higher $pp$ collider) [55].

### 4. Summary

We proposed a new way to introduce a relativistic deformation of the kinematics of SR. In this proposal, the deformation only applies to the interaction of elementary particles. What is elementary depends on the energy (and associated length) scale of the interaction considered. In this way, we can restrict the effects of the deformation to those particles which we have not been able to identify as composite particles. Thus, we obtain a new perspective of DSR where the different potential inconsistencies (soccer-ball and spectator problems, consistency with tests of locality) are automatically solved.

In order to explore the consequences of this proposal, we have to incorporate the effects of DSR in a model for the interaction of particles that gives a generalization of the result of the perturbative treatment in RQFT for cross sections and decay widths compatible with the deformed relativistic invariance of DSR. This will be the subject of future work.

**Author Contributions:** Writing—original draft preparation, review and editing, J.M.C., J.L.C., J.J.R. and M.A.R. All authors contributed equally to the present work. All authors have read and agreed to the published version of the manuscript.

**Funding:** This work is supported by Spanish grants PID2021-126078NB-C21 (FEDER/AEI) and DGIID-DGA no. 2020-E21-17R. The work of M.A.R. was supported by MICIU/AEI/FSE (FPI grant PRE2019-089024). This work has been partially supported by Agencia Estatal de Investigación (Spain) under grant PID2019-106802GB-I00/AEI/10.13039/501100011033, by the Regional Government of Castilla y León (Junta de Castilla y León, Spain) and by the Spanish Ministry of Science and Innovation MICIN and the European Union NextGenerationEU/PRTR. The authors would like to acknowledge the contribution of the COST Action CA18108 "Quantum gravity phenomenology in the multi-messenger approach".

**Data Availability Statement:** No new data were created or analyzed in this study. Data sharing is not applicable to this article.

**Conflicts of Interest:** The authors declare no conflict of interest.

### Appendix A. Modified Composition of Momenta

A relativistic deformation of SR in which the dispersion relation and Lorentz transformations in the one-particle system are not deformed is provided within the mathematical framework of Hopf algebras by the classical basis of $\kappa$-Poincaré [35]. The coproduct of the momentum operators defines a composition law of momenta

$$(p \oplus q)_0 \;=\; p_0\,\Pi(q) + \Pi^{-1}(p)\left(q_0 + \frac{\vec{p}\cdot\vec{q}}{\Lambda}\right), \qquad (p \oplus q)_i \;=\; p_i\,\Pi(q) + q_i\,, \qquad \text{(A1)}$$

with

$$\Pi(k) \;=\; \frac{k_0}{\Lambda} + \sqrt{1 + \frac{k_0^2 - \vec{k}^2}{\Lambda^2}}\,, \qquad \Pi^{-1}(k) \;=\; \left(\sqrt{1 + \frac{k_0^2 - \vec{k}^2}{\Lambda^2}} - \frac{k_0}{\Lambda}\right)\left(1 - \frac{\vec{k}^2}{\Lambda^2}\right)^{-1}, \text{ (A2)}$$

where $\Lambda$ plays the role of the high-energy scale deforming the kinematics. From this composition law of momenta, one defines the antipode (inverse element of the composition) $\hat{p}$ of a momentum $p$ satisfying $(\hat{p} \oplus p) = (p \oplus \hat{p}) = 0$. For the classical basis, it reads

$$\hat{p}_0 \;=\; -p_0 + \frac{\vec{p}^2}{\Lambda}\,\Pi^{-1}(p)\,, \qquad \hat{p}_i \;=\; -p_i\,\Pi^{-1}(p)\,. \qquad \text{(A3)}$$

The nontrivial coproduct of Lorentz generators provides a Lorentz transformation of the two-particle system compatible with the linear Lorentz transformation of the total momentum defined by the noncommutative composition law (A1).

This relativistic deformation of SR can also be obtained from a geometric perspective [56]. In fact, the step from SR to DSR can be understood as going from a flat Minkowski momentum space to a curved momentum space [57,58]. If one considers the metric [59]

$$g_{\mu\nu}(k) = \eta_{\mu\nu} + \frac{k_\mu k_\nu}{\Lambda^2} \qquad (A4)$$

in a de Sitter maximally symmetric curved momentum space as a deformation of the flat Minkowski metric $\eta_{\mu\nu}$, one can see that the isometries of this metric reproduce the kinematics of the classical basis of $\kappa$-Poincaré. Note that, in particular, the absence of a deformation of the dispersion relation is compatible with the usual interpretation of the dispersion relation as given by (a function of the square of) the geometric distance from the origin ($k = 0$) to a point $k = p$ with the previous metric. On the other hand, the composition law of momenta (A1) is the result of a translation in momentum space with the components of one of the two momentum variables identified as the set of parameters of the translation.

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
