# Peer review of "A New Perspective on Doubly Special Relativity"

_universe, doi:10.3390/universe9030150_

Round 1

Reviewer 2 Report

The paper ¨A new perspective on Doubly Special Relativity” brings new insights to a possible different interpretation of DSR proposals. Without focusing in on or other particular model, the novel perspective of the authors consists, basically, of considering the deformation to affect only elementary particles. We have, in this current view some interesting consequences, namely,

1. Deviations of the energy-momentum relation of a single free particle is not attainable. There is but only one way have observable effects – the energy of interacting particles must be close of the energy parameter of the corresponding deformation.

2. The so-called spectator and also the soccer-ball problems are automatically solved.

I have only two points, to be considered by the authors. First of all, the second paragraph in page 3, line 100/101 seems to leverage DSR as a theory when they say

“we conclude that the energy scale \Lambda of DSR should be larger than a TeV.’’

Perhaps a statement at the end such as

“if, of course, is in fact correct.’’

could improve the scientific character of the present manuscript.

Secondly, there is a long-standing problem concerning DSR proposals, which is the lack of a space-time consistent approach, since they are always formulated in the energy-momentum space. How the new perspective proposed here is supposed to shed some light in this issue?

Despite the two points above, to be addressed by the authors and due to my initial exposition, I believe the paper is almost ready for publication.

Reviewer 3 Report

"A new perspective on Doubly Special Relativity" is a good manuscript and I enjoyed reading it. I appreciate the authors for their beautiful paper. But, it will be better that the authors provide more mathematical formulation for their perspective. I think this manuscript can be published in the Journal of the Universe after improvement.  

Review:   The authors haven't provided any serious mathematical formulation of their new perspective. By mentioning some problems of the standard DSR such as the “soccer-ball” and  “spectator problem”, they have suggested restricting DSR theory only to the interactions between elementary fundamental particles.   For introducing this new perspective,  I think the authors should introduce a new modified field theory first. For example, what will be the modified Dirac equation for two interacting electrons? Without a mathematical formalism, the manuscript will be decreased to a philosophical paper. It is better that the authors provide at least one or two examples of the modified Lagrangian for the interacting elementary particles.

Also, I am wondering what will be the form of the Magueijo-Smolin transformations ( hep-th/0112090) in this perspective?    

In the sentence:"while in the new perspective proposed in this work the deformation is introduced in the relation between moments associated to product of quantum fields.",  So we can ask: Is there any formula for this?  

END      

Reviewer 4 Report

This paper presents an interesting discussion on the energy-momentum relations of particles in the doubly special relativity theory. I think the discussions and conclusions are useful contribution to this field. I recommend the publication of this paper in Universe.

Round 2

Reviewer 1 Report

In the revised version of the manuscript the authors address the questions that were raised in the previous report. For this reason we believe that the paper has improved and it is in principle publishable.

Reviewer 3 Report

I think the paper " A new perspective on Doubly Special Relativity" is ready for publication. I advise publication of this paper in the present form in the Journal of the Universe.  It can initiate useful discussions on the interpretation of  the DSR theory in the phenomenology of the quantum gravity.